# Influence of Phase Change on Parthenogenesis in the Migratory Locust: A Behavioral Analysis

**DOI:** 10.3390/insects16090878

**Published:** 2025-08-23

**Authors:** Rongjing Li, Xuelei Wei, Liwei Zhang

**Affiliations:** College of Grassland Science and Technology, China Agricultural University, Beijing 100193, China; lirongjing2004@126.com (R.L.); weixuelei1123@163.com (X.W.)

**Keywords:** migratory locust, parthenogenesis, density, phase polyphenism, fertilization

## Abstract

Our study has demonstrated the differences in parthenogenesis between gregarious and solitary locusts, as well as their disparities compared to sexual reproduction. We evaluated these aspects from the perspectives of oviposition dynamics, number of oviposition events, quantity of oviposition, and hatching rates. Furthermore, we discovered that density plays a significant regulatory role in locust reproductive strategies. At the behavioral level, we revealed the parthenogenetic strategies of locusts, deepening the understanding of the phenomenon of parthenogenesis and providing theoretical support for the formulation of future locust control strategies.

## 1. Introduction

In the animal kingdom, some female individuals of certain species are capable of producing eggs that can develop into new individuals without fertilization, a phenomenon known as parthenogenesis [1]. Parthenogenesis can be classified into two types: facultative parthenogenesis and obligate parthenogenesis. The fundamental distinction between these two types lies in whether the products of meiosis undergo fusion phenomenon during the process of parthenogenesis. In facultative parthenogenesis, the sister chromatid experiences separation through meiosis, and the resulting haploid cells (egg cells and polar bodies) fuse to restore the diploid state [2]. Conversely, in obligate parthenogenesis, this process is replaced by a modified meiosis that maintains ploidy [3]. Some scholars argue that facultative parthenogenesis is not true parthenogenesis, as it retains some characteristics of sexual reproduction and allows organisms to switch between sexual and asexual parthenogenetic reproduction [4]. In general, parthenogenesis is rare among vertebrates, but it is frequently observed in many invertebrate groups [5], such as insects. Obligate parthenogenesis, found in around 100 vertebrate species and 1000 invertebrate species, is relatively rare [6]. Conversely, facultative parthenogenesis, where females can reproduce both sexually and parthenogenetically, occurs in around 20% of all animal species employing haplodiploid sex determination. In such species, whilst females are produced sexually, deriving from eggs that have been fertilized by sperm, males are produced asexually, deriving from unfertilized eggs, in a process termed “arrhenotoky” [7].

Parthenogenesis is an important reproductive strategy of insects and is of great significance to their survival against environmental stress. From an evolutionary perspective, parthenogenesis is advantageous for insect populations, particularly when the number of male individuals is scarce. In such cases, female individuals can utilize parthenogenesis to ensure the continuation of the population [8]. It has been reported that several species of locusts are capable of parthenogenesis. The desert locust (*Schistocerca gregaria*) exhibits parthenogenesis, and the offspring produced passed over four generations. This study reported an average hatching rate of 25%, compared to around 75% following sexual reproduction [9]. Similarly, the migratory locust (*Locusta migratoria*) possesses the ability for parthenogenesis [10]. On the other hand, it has shown that the sexual reproduction capacity of solitary locusts is significantly higher than that of gregarious locusts [11]. However, there is limited knowledge about the parthenogenetic differences between solitary and gregarious phases in migratory locusts. Thus, we designed a series of experiments based on the phase change and the effects of different population densities on parthenogenesis in the migratory locust (*Locusta migratoria*) aiming to investigate the differences between parthenogenesis and sexual reproduction, as well as the variations in parthenogenesis between gregarious and solitary locusts. This research seeks to deepen our understanding of parthenogenesis and provide potential support for the control of migratory locust (*Locusta migratoria*) outbreaks.

## 2. Materials and Methods

### 2.1. Establishment of Insect Populations

The eggs of the migratory locust (*Locusta migratoria*) used in this experiment were sourced from the Insect Behavior and Pest Biocontrol Laboratory at China Agricultural University as part of a long-term laboratory colony. The locusts were reared from hatching onward in gregarious nymph rearing cages with a density of about 200 individuals per cage (cylindrical containers with a diameter of approximately 12.4 cm and a length of about 30 cm, approximately 0.055 individuals/cm^3^). The rearing conditions were maintained as follows: temperature of 28–32 °C, humidity of 30–60%, and a light cycle of 18 h light and 6 h dark (L:D = 18:6). The locusts were fed daily with an adequate amount of fresh wheat seedlings grown in the laboratory. After eclosion, the adults were transferred to the corresponding adult rearing cages with defined density level.

### 2.2. Density Setup (For All Groups Used in This Study)

GG Group (*Gregarious phase in both nymph and adult stages*): The nymphs were reared at a density of 200 individuals per cage (approximately 0.055 individuals/cm^3^) in gregarious nymph rearing cages. They were fed fresh wheat seedlings daily until they molted into adults. After eclosion, 26 female adults were transferred to well-ventilated small gregarious adult rearing cages (18 cm × 18 cm × 18 cm), with a density of approximately 0.0045 individuals/cm^3^. This GG group indicates parthenogenetic reproduction of gregarious locusts.

SS Group (*Solitary phase in both nymph and adult stages*): The nymphs were reared at a density of a single individual per box (approximately 0.0013 individual/cm^3^) in well-ventilated solitary locust-rearing boxes (cubic containers with a height of approximately 12 cm and length and width of about 8 cm each). After eclosion, male and female adults were reared separately in solitary locust-rearing boxes at a density of approximately 0.0013 individuals/cm^3^. This SS group indicates parthenogenetic reproduction of solitary locusts.

B0 group (Sexually fertilized GG group): The nymphs were reared at a density of 200 individuals per cage in well-ventilated gregarious nymph rearing cages. After eclosion, 40 male and 40 female adults were transferred to well-ventilated large gregarious adult rearing cages (30 cm × 30 cm × 30 cm) to allow normal mating. This group indicates sexual fertilization.

To demonstrate the plasticity of rearing density in adult stage on parthenogenesis, the gregarious nymphs were isolated since eclosion in solitary rearing cages. Briefly, the gregarious nymphs were reared at three density levels: low level (40 individuals per cage, approximately 0.011 individuals/cm^3^), medium level (100 individuals per cage, approximately 0.028 individuals/cm^3^), and high level (250 individuals per cage, approximately 0.069 individuals/cm^3^) in well-ventilated gregarious nymph rearing cages (cylindrical containers with a diameter of approximately 12.4 cm and a length of about 30 cm). They were fed fresh wheat seedlings daily until they molted into adults. After eclosion, a single female adult from each density level were reared in solitary-rearing boxes at a density of approximately 0.0013 individuals/cm^3^, termed as GSL Group (*low-density gregarious nymph isolated into solitary adult*), GSM Group (*medium-density gregarious nymph isolated into solitary adult*) and GSH Group (*high-density gregarious nymph isolated into solitary adult*), respectively. These three groups indicate parthenogenetic reproduction of isolated gregarious locusts. Table 1 below summarizes the feeding conditions of each group

### 2.3. Parthenogenesis Oviposition Setup

After eclosion, all individuals were maintained under the optimal rearing conditions until their death. At 5–7 days after eclosion, when the locusts were reaching sexual maturity, cylindrical plastic oviposition boxes (7 cm in diameter, 6 cm in height) filled with moist vermiculite were placed in the rearing cages to collect eggs. These boxes were designed to fit into a designated circular notch in the cage bottom, with their upper rims fixed to ensure they fully covered the notch without affecting the individuals’ living space. Every 24 h, the oviposition box was checked and changed. The eggs were disinfected by spraying with 75% ethanol and placed in cylindrical plastic incubation boxes (5.5 cm in diameter, 3 cm in height), covered with moist vermiculite that had been sterilized for hatching at 30 °C. The oviposition, hatching, and mortality were recorded throughout the experiments every day. After 30 days, when no further hatching occurred, the egg pods were carefully opened to count the remaining unhatched eggs. Severely moldy eggs were omitted from the results. The oviposition dynamics, accumulative hatching rate, and hatching curves were calculated.

To conduct a detailed analysis of reproductive changes in migratory locusts, we categorized each individual into different oviposition stages (the GG group, which was collectively reared and thus individuals could not be tracked individually, was categorized as a group) called “spawning stage”: 0–50% of the eggs laid by each individual were considered to be from the early stage; 50–75% were considered to be from the middle stage; and 75–100% were considered to be from the late stage (for the GG group, these categories were applied to the entire group’s oviposition output: 0–50%, 50–75%, and 75–100%). This approach was used to investigate changes in reproductive patterns throughout the migratory locust life cycle.

We have created Figure 1: Temporal Oviposition Dynamics Across Groups, where the y-axis labeled “Average number of pods produced per individual” indicates the average number of egg pods produced per individual during the day. We have also developed Figure 2: Comparison of Oviposition Fecundity Between Parthenogenesis and Sexual Reproduction in Three Groups. Specifically, in Figure 2a, the y-axis “Average number of pods produced per individual” represents the average number of egg pods produced per individual during our experiment; in Figure 2b, the y-axis “Average number of eggs per pod” denotes the average number of eggs per egg pod; and in Figure 2c, the y-axis “Average number of eggs produced per individual” refers to the average number of eggs produced per individual during our experiment. Additionally, we have produced Figure 3: Comparison of Hatching Rate and Temporal Dynamics Across Groups, with the y-axis “Rate%” signifying the hatching rate. Finally, we have created Figure 4: Temporal Hatching Dynamics in Three Groups, where the y-axis “Number of daily hatches/Number of total eggs” indicates the proportion of the hatched eggs during the day to the total number of eggs.

### 2.4. Data Fitting

Oviposition and hatching dynamics from all density levels were fitted. Data were fitted using GraphPad Prism 9.5.0 software for Gaussian fitting and the goodness-of-fit was annotated below the corresponding figure.

### 2.5. Statistical Analysis

Two-tailed Student′s *t* test were used for two-group comparisons. For multiple-group comparison, one-way ANOVA followed by Tukey′s test was applied. The differences were considered significant when *p* < 0.05. All statistical analyses were performed by GraphPad Prism 9.5.0 software. Data are represented as mean ± SEM. All statistical details can be found in the according figure legends.

## 3. Results

### 3.1. The Impact of Phase Change on the Oviposition Dynamics of Parthenogenesis

The migratory locust exhibits density-dependent phase polyphenism, or phase change. High-density induces gregarious phase, and low-density forms solitary phase. Huge physiological and behavioral differences occur between two phases, such as reproduction, immunity, development, and chemical communications. We want to figure out whether parthenogenetic reproduction differs between two phases, similar to sexual reproduction. Firstly, the temporal dynamics of oviposition over 50 days were compared. It can be observed that the oviposition peaks for the SS (parthenogenetic solitary group), GG (parthenogenetic gregarious group), and B0 groups (sexual GG group) are concentrated between 20 and 30 days (Figure 1a). GG group peaked earlier and laid the least number of egg pods compared to the other two groups. Interestingly, GG group peaked earlier and laid the least number of egg pods, while SS laid the most. Since population density level determines the phase state of the migratory locusts, we further manipulated the adult density from gregarious nymphs and checked the parthenogenesis dynamics. When gregarious nymphs were separated into three density levels (see Materials and Methods), peak laying by the resulting unfertilized females concentrated at around 15 days, but lower-density gregarious nymphs led to laying over a longer time period, with more egg pods than with any of the others including GG (Figure 1b: GSL). Medium-density gregarious nymphs led to laying over a shorter period, while high-density gregarious nymphs led to laying of only a few egg pods over a very short period (Figure 1b). Moreover, parthenogenetic fecundity increased in GSL and GSM groups, in comparison with GG group. Thus, increasing density in adults might decrease the oviposition capacity with sexual fertilization, which is consistent with the parthenogenetic difference observed between solitary and gregarious locusts.

### 3.2. The Impact of Phase Change on the Parthenogenetic Fecundity

Next, we compared the parthenogenetic fecundity among three groups by quantifying the number of egg pods and egg numbers in each pod. Solitary females (SS) laid the highest number of egg pods (Figure 2a) but had the least number of eggs per pod (Figure 2b). In contrast, fertilized gregarious females (B0) laid fewer pods (Figure 2a) but had the most eggs per pod (Figure 2b). The two overall results were that SS and B0 laid a similar number of eggs (Figure 2c). Gregarious females that were unfertilized (GG) laid only a few pods (Figure 2a). Some of the pods had a similar number of eggs per pod as the average for fertilized gregarious females (B0), while others had very few eggs (Figure 2b), resulting in far fewer eggs per parthogenetic female (Figure 2c). Meanwhile, GG group produced the least egg pods with the least eggs inside, resulting few eggs laid by each parthenogenetic female. Thus, density level might re-shape the development of female oocyte to change the number of mature eggs in females.

### 3.3. The Impact of Phase Change on the Hatching Rate and Hatching Dynamics of Parthenogenesis

In general, parthenogenetic reproduction leads to low hatchlings; however, whether phase change would re-shape the parthenogenetic success is unclear. To answer this, we quantified the hatchling rates among groups and found that the fertilized B0 group hatched more than 80%, as expected (Figure 3a). Parthenogenetic eggs from both GG and SS groups hatched rarely, but the hatchling rate of SS group was significantly higher than GG group (19% vs. 3%). This indicates that crowding females would suppress parthenogenesis via certain signaling, which is inhibited in solitary females. We also compared the hatchling dynamics in three groups. In B0 and GG groups, eggs laid at different times in the female life-stage did not have any difference in hatchling rate, thus indicating that all eggs share similar developmental success (Figure 3b,d). Conversely, the SS group exhibited a gradual increase in egg hatching rate throughout the life cycle (Figure 3c), observed with an enhanced parthenogenesis success in eggs that were laid in the late life-stage. Together, hatching rate and dynamics are highly associated with phase polyphenism.

### 3.4. The Impact of Phase Change on the Hatching Peaks of Parthenogenesis

In the end, we compared the hatching peaks among groups. It can be observed that the hatching peak times of all three groups are similar. The hatching peaks of the B0 and SS groups were similar, with the GG group slightly delayed (Figure 4). However, the peak values differ significantly (daily hatchling number: B0 > SS > GG). Additionally, the onset of hatching occurs earlier in the B0 group compared to the SS group, which in turn occurs earlier than the GG group. These data demonstrated that sexual fertilization accelerates hatching in comparison to parthenogenetic reproduction.

## 4. Discussion

Locusts change reversibly between solitary and gregarious phases that differ dramatically in appearance, general physiology, brain function and structure, and behavior. For example, adipokinetic response (= lipid mobilization induced by adipokinetic hormones) is more intense in crowded than in isolated adults of *Locusta migratoria migratorioides* [12]. Long-term gregarious desert locusts have a smaller body size, but their brain is substantially larger—about 30%—than that of long-term solitary locusts. In addition, the relative distribution of brain regions differs between the two phases. Solitary locusts invest more in lower-level sensory processing, reflected by their relatively large primary olfactory and visual neuropils. In contrast, the larger brains of gregarious locusts are more dedicated to the integration of sensory cues in higher-level processing regions, which is thought to support their lifestyle as generalist foragers in dense, migratory swarms where competition among group members is high [13,14]. Solitary locusts actively avoid contact with other locusts, but gregarious locusts may live in vast, migrating swarms dominated by competition for scarce resources and interactions with other locusts. Different phase traits change at different rates; some behaviors take just a few hours, coloration takes a lifetime and the muscles and skeleton take several generations [15]. In the initial stages of this experiment, we attempted to isolate the GG group after eclosion to observe their parthenogenesis capacity. However, the results were inconsistent upon multiple repetitions. Eventually, we discovered that this phenomenon was related to the population density of nymphal locusts, leading to the subsequent experimental setup by quantifying three density levels of gregarious nymphs (GSL/M/H). Based on our observations, we speculate that the parthenogenesis in locusts may fall under the category of facultative parthenogenesis, as both solitary and gregarious locusts are capable of normal mating, albeit with differing mating strategies [16]. A particularly interesting observation in this experiment was that solitary locusts exhibited more frequent oviposition events with smaller egg numbers in each pod, whereas gregarious locusts had fewer oviposition events with larger egg numbers in each pod. This difference may be linked to the reproductive strategies between phases; solitary locusts may be more proficient and cautious in parthenogenesis to guide offspring survival. In natural conditions, solitary locusts may struggle to find mates, thus resorting to more frequent but smaller-scale events. In contrast, gregarious locusts have greater opportunities to find mates due to their densely populated environment, and thus may delay oviposition to wait for males, which results in a significant reduction in oviposition frequency. The distinct hatching rates of parthenogenetic eggs between two phases further support this hypothesis.

Another intriguing observation was that the hatching rates of solitary parthenogenetic eggs increased with the advancing life stages, but not in gregarious eggs. This pattern may be part of making certain some successful reproduction occurs through investing more energy into parthenogenic reproduction in later stages when the females are getting older and nearer death. We found that solitary-phase locusts typically die soon (within 3.5 days) after their final oviposition event, which may be related to this reproductive adjustment. This could be an example of terminal investment, as indicated by a decrease in the expectation of future reproduction, resulting in an increase in reproductive investment to ensure the continuation of the population. Although terminal investment is often treated as a static strategy, the level at which a cue of decreased future reproduction is sufficient to trigger increased current reproductive effort (i.e., the terminal investment threshold) may depend on the context, including the internal state of the organism or its current external environment, independent of the cue that triggers a shift in reproductive investment [17]. Additionally, for both B0 and SS, hatching is a normal distribution about the mean so that both have a mean + standard error. B0 may have a slightly larger S.E. so that hatching begins earlier and continues later, but it may just be that there are more hatchings, so some begin earlier.

As previously discussed, parthenogenesis is an important reproductive strategy in insects, yet its manifestation varies across species. *Drosophila mangabeirai* is the only species known to exhibit obligate parthenogenesis in genus *Drosophila* [18], while most other species can engage in varying degrees of facultative parthenogenesis [19]. Additionally, the hatching rates of parthenogenetic eggs are generally lower than those of sexually produced eggs [20], a pattern consistent with our findings in locusts. Another example within the Orthoptera order, *Warramaba virgo*, exhibits obligate parthenogenesis and originated from hybridization between *Warramaba whitei* and *Warramaba flavolineata*. Interestingly, this species has not exhibited inbreeding depression, and studies suggest that its oviposition quantity slightly exceeds that of certain populations of *W. whitei* [21]. However, there is no available research on the hatching rates of parthenogenetic eggs in this species. This conflicts with the results from our study, as based on our observations, the offspring produced by parthenogenesis in the migratory locust have extremely low survival rates and hardly undergo metamorphosis into adults. Therefore, we believe that parthenogenesis in the migratory locust leads to certain genetic defects in the offspring. In social Hymenoptera insects, such as bees and ants, parthenogenesis is a normal component of their life cycle, regardless of whether individuals are solitary or social. Male offspring develop from unfertilized eggs through parthenogenesis, making them haploid, while females develop from fertilized eggs. In exceptional cases, females can also produce female offspring through parthenogenesis [22,23]. However, based on our observations and previous studies, the parthenogenesis of the Oriental migratory locust is contrary to this; the offspring consist solely of females, with no males produced [24]. Within a single colony, different reproductive modes (sexual and asexual) can coexist among queens, enabling them to benefit from the advantages of both sexual and clonal reproduction. By using alternative modes of reproduction for the queen and worker castes, queens can increase the transmission rate of their genes to their reproductive female offspring while maintaining genetic diversity and social cohesion in the worker population [25]. In the cockroach species *Reticulitermes speratus*, the hatching rates of parthenogenetic and sexually produced eggs are similar, although parthenogenetic eggs exhibit delayed initial hatching [26]. In our study, the hatching rates of parthenogenetic eggs from both solitary and gregarious locusts were significantly lower than those of sexually produced eggs, and the parthenogenetic eggs also showed slightly delayed initial hatching. Parthenogenesis is also observed in Lepidoptera, such as the potato tuber moth (*Phthorimaea operculella*), where parthenogenetic eggs exhibit lower hatching rates, higher larval mortality, and shorter lifespans compared to sexually produced eggs [27]. This suggests that parthenogenesis in this organism may have developmental defects, consistent with our observations in locusts. Additionally, insects exhibit a unique form of parthenogenesis known as viviparity, which is primarily observed in aphids. Most modern aphids possess two apparent evolutionary novelties: cyclical parthenogenesis (a life cycle with both sexual and asexual phases) and viviparity (internal development and live birth of progeny) in their asexual phase [28]. This special phenomenon is unique to aphids and rarely reported in the Oriental migratory locust or other insects. Recent studies on stick insects have highlighted their remarkable reproductive diversity. A striking aspect of the biology of stick insects is the widespread occurrence of parthenogenesis, including rare, spontaneous events in sexual species, facultative parthenogenesis as well as obligately parthenogenetic species [29]. Some researchers suggest that facultative parthenogenesis in stick insects may often be a transient strategy that could eventually be replaced by obligate reproductive modes (either sexual or parthenogenetic) [30]. This is an extremely special type of reproductive strategy, which may become a future research hotspot.

Additionally, the Drosophila embryos possessed maternally derived cell regulators that could trigger the cleavage of early embryonic cells. As the maternal protein was insufficient for further embryonic development, mRNA stored during oogenesis was used to translate the cyclic regulators during the synthesis of the cleavage process, and the early embryonic development of eggs was inseparable from that of the accumulation of oocytes in the ovary. Furthermore, the cleavage rate of newly produced eggs might have been higher due to the existence of the maternal mRNA protein, with the degradation of maternal mRNA, the rate of cleavage slowed and the egg initiated the nucleus genome to encode for new protein synthesis. Consistent with the research in Drosophila, two types of termites show similar situations of embryonic development termination; the cleavage rate of *R. aculabialis* slowed gradually around the fourth day. The egg development of *R. flaviceps* ceased in the blastoderm formation after approximately 8–10 days postoviposition. According to the author that after the degradation of the maternal mRNA, the genome of *R. flaviceps* was unable to begin encoding for new proteins, which would stop any further development of the embryo [31,32]. We hypothesize that a similar mechanism exists in locusts; undeveloped locust eggs may result from the absence of maternally derived cell regulators, while development-arrested locust eggs may be due to the degradation of maternal mRNA and the failure to initiate transcription. Interestingly, previous studies have shown that forcing fruit flies to undergo parthenogenesis repeatedly leads to a significant increase in the developmental rate of locust eggs and the survival rate of parthenogenetic offspring after multiple generations of parthenogenesis [33]. This may suggest the presence of specific signaling pathways regulating parthenogenesis, which can be selectively purified to enhance the success of parthenogenetic reproduction.

In summary, the parthenogenesis strategy of locusts shares similarities and differences with those of other insects, with the differences primarily reflecting their strong phenotypic plasticity and traits regulated by population density. While this study provides valuable insights into the mechanisms and ecological implications of parthenogenesis in locusts, several limitations remain. For instance, the hormonal regulation underlying parthenogenetic behavior and the genetic basis of parthenogenesis in locusts remain poorly understood. Future research should focus on addressing these gaps to achieve a comprehensive understanding of the mechanisms and evolutionary significance of parthenogenesis in locusts.

## 5. Conclusions

Our experimental results revealed that gregarious locusts exhibit weaker parthenogenetic capacity compared to solitary locusts. However, the parthenogenetic strategy of solitary locusts includes frequent, smaller oviposition events. Specifically, the average number of oviposition events per solitary locust was 2.6 times higher than that of gregarious locusts. Furthermore, while the average number of eggs laid per solitary locust in parthenogenesis was comparable to that of sexual reproduction, it reached approximately 2.7 times that of gregarious locusts in parthenogenesis. There were also significant differences in hatching rates; sexual reproduction in locusts achieved a hatching rate of 85%, while parthenogenesis in solitary locusts was approximately 15%, and parthenogenesis in gregarious locusts was only 3%.

Interestingly, solitary parthenogenesis exhibited a phenomenon where hatching rates increased with the progression of the locust’s life cycle, with eggs laid during the final 25% of the locust’s life achieving an average hatching rate three times higher than those laid during the first 50%. Although gregarious parthenogenesis showed a similar trend, it was not statistically significant, and sexual reproduction exhibited no such phenomenon. The non-significance with gregarious locusts may reflect the far fewer eggs hatching. Additionally, we observed that the parthenogenetic capacity of gregarious locusts decreased with increasing juvenile densities, demonstrating a strong relationship between parthenogenesis and density changes in locusts, as well as a high degree of phenotypic plasticity.

## Figures and Tables

**Figure 1 insects-16-00878-f001:**
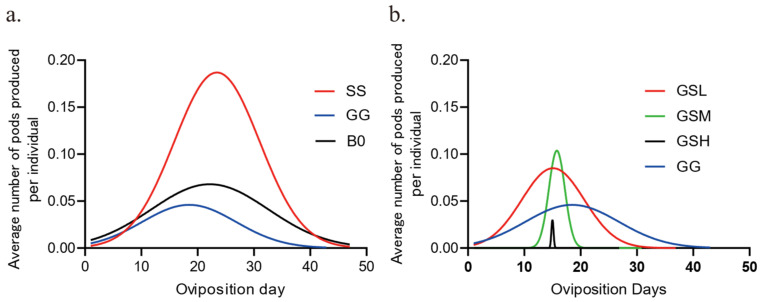
Temporal oviposition dynamics across groups. (**a**). Fitting curves of oviposition dynamics in three groups. n = 29 for SS, R^2^ = 0.5855; n = 26 for GG, R^2^ = 0.3471; n = 40 for B0; R^2^ = 0.2327. (**b**). Fitting curves for oviposition dynamics in isolated gregarious adults. n = 37 for GSL, R^2^ = 0.4138; n = 30 for GSM, R^2^ = 0.6178; n = 30 for GSH, R^2^ = 0.4600; n = 26 for GG, R^2^ = 0.3471.

**Figure 2 insects-16-00878-f002:**
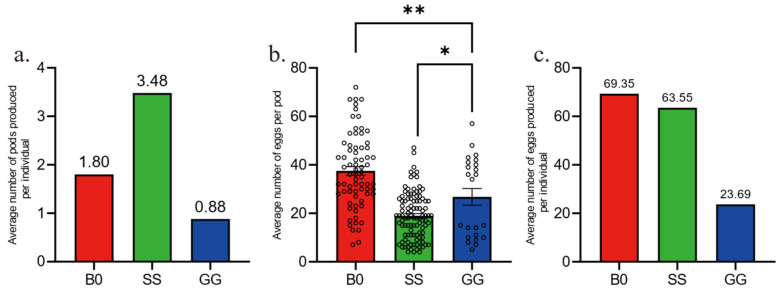
Comparison of oviposition fecundity between parthenogenesis and sexual reproduction in three groups. (**a**). Average number of egg pods per female in three groups. n = 72 for B0, n = 101 for SS, n = 23 for GG. (**b**). Average number of eggs per pod in three groups. n = 71 for B0, n = 83 for SS, n = 23 for GG. (**c**). Average number of eggs per female in three groups. n = 71 for B0, n = 83 for SS, n = 23 for GG. The data in (**b**) are shown as the mean ± SEM. One-way ANOVA followed by Tukey’s test, * *p* < 0.05, ** *p* < 0.01. Empty circles represents individual replicates.

**Figure 3 insects-16-00878-f003:**
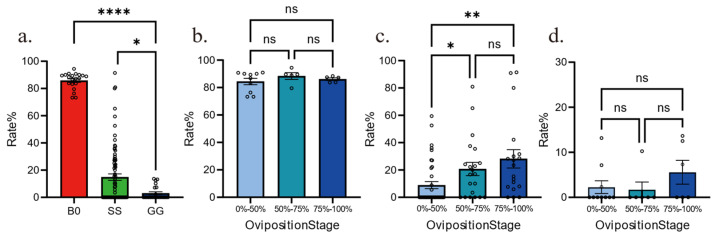
Comparison of hatching rate and temporal dynamics across groups. (**a**). Hatching rate among three groups. n = 20 for B0, n = 83 for SS, n = 23 for GG. (**b**). Hatching rates of eggs from different life stages for B0 group. (**c**). Hatching rates of eggs from different life stages for SS group. (**d**). Hatching rates of eggs from different life stages for GG group. All data are shown as the mean ± SEM. One-way ANOVA followed by Tukey’s test, ns, *p* > 0.05, * *p* < 0.05, ** *p* < 0.01, **** *p* < 0.0001. Empty circles represents individual replicates.

**Figure 4 insects-16-00878-f004:**
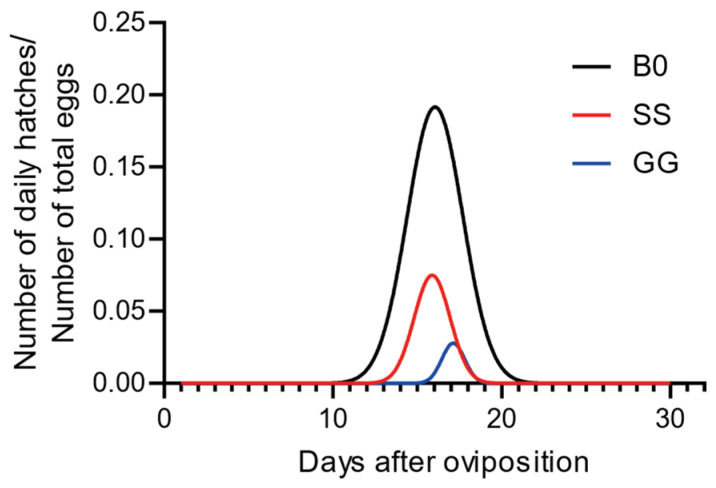
Temporal hatching dynamics in three groups. n = 20 for B0, R^2^ = 0.8556; n = 83 for SS, R^2^ = 0.9788; n = 23 for GG, R^2^ = 0.9193.

**Table 1 insects-16-00878-t001:** Rearing conditions for each experimental group.

Group	Nymphs	Adults	Reproduction
SS	Solitary individual	Solitary individual	Unfertilized
GG	Gregarious	Gregarious	Unfertilized
B0	Gregarious	Gregarious	Fertilized
GSL	Low-density Gregarious	Solitary individual	Unfertilized
GSM	Medium-density Gregarious	Solitary individual	Unfertilized
GSH	High-density Gregarious	Solitary individual	Unfertilized

## Data Availability

The data presented in this study are available on request from the corresponding author due to privacy reasons.

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
