# Peer review of "Influence of Phase Change on Parthenogenesis in the Migratory Locust: A Behavioral Analysis"

_insects, 2025, doi:10.3390/insects16090878_

Round 1

Reviewer 1 Report

Comments and Suggestions for Authors

Li, et al., described the parthenogenesis in the migratory locust, and found the difference of parthenogenesis capacity between solitary and gregarious phases. In detail, solitary females laid more frequently and more egg pods than gregarious counterparts without sexual fertilization, but the egg-laying peak in solitary females was little late than gregarious females, temporally. Solitary locusts exhibit a higher parthenogenesis capacity compared to gregarious locusts, as evidenced by greater total oviposition quantity and higher hatching rates. The parthenogenetic solitary eggs resulted in lower hatching rate than eggs upon sexual fertilization, as expected. The parthenogenesis features between two phases were density-dependent, as shown that gregarious locusts, when isolated after eclosion, exhibited increased parthenogenetic capacity, depending on their juvenile density. Overall, this is a simple, interesting, and straightforward study, clearly demonstrating that solitary females can produce offspring via parthenogenesis. Here are some of my comments.

1. The author has provided parthenogenetic phenotype between two locust phases, but they didn’t discuss the potential mechanism underlying this interesting phenomenon, for example, what happened during embryogenesis. I recommend providing critical discussion, to direct future exploration in molecular and cellular mechanism.

2. I’m curious about the offspring sex of solitary parthenogenesis, and whether they can live over whole life to make normal courtship with males? Or whether they can continue produce parthenogenetic nymphs to finish kind of self-cross?

Author Response

Reviewer2:

Li, et al., described the parthenogenesis in the migratory locust, and found the difference of parthenogenesis capacity between solitary and gregarious phases. In detail, solitary females laid more frequently and more egg pods than gregarious counterparts without sexual fertilization, but the egg-laying peak in solitary females was little late than gregarious females, temporally. Solitary locusts exhibit a higher parthenogenesis capacity compared to gregarious locusts, as evidenced by greater total oviposition quantity and higher hatching rates. The parthenogenetic solitary eggs resulted in lower hatching rate than eggs upon sexual fertilization, as expected. The parthenogenesis features between two phases were density-dependent, as shown that gregarious locusts, when isolated after eclosion, exhibited increased parthenogenetic capacity, depending on their juvenile density. Overall, this is a simple, interesting, and straightforward study, clearly demonstrating that solitary females can produce offspring via parthenogenesis. Here are some of my comments.

Response:

We thank the reviewer for these comments.

The author has provided parthenogenetic phenotype between two locust phases, but they didn’t discuss the potential mechanism underlying this interesting phenomenon, for example, what happened during embryogenesis. I recommend providing critical discussion, to direct future exploration in molecular and cellular mechanism.

Response:

Thanks. This provides valuable suggestions for our future research direction. And we discussed our results from the perspective of embryonic development. We speculate that the reasons for the differences in embryonic development between parthenogenesis and sexual reproduction may have similarities with those in fruit flies and termites, and we have discussed this in the discussion section

I’m curious about the offspring sex of solitary parthenogenesis, and whether they can live over whole life to make normal courtship with males? Or whether they can continue produce parthenogenetic nymphs to finish kind of self-cross?

Response:

Thanks. Based on our observations and previous studies, the offspring produced by parthenogenesis in the Migratory Locust are all female, but with extremely low survival rates. Consequently, we only observed that the offspring from parthenogenesis can undergo parthenogenesis themselves, while there is a lack of experimental evidence for sexual reproduction.

Reviewer 2 Report

Comments and Suggestions for Authors

Overall Evaluation:

This is a very interesting and valuable study. The authors investigate how population density influences parthenogenesis in locusts, a species known for striking density-dependent phenotypic plasticity. Surprisingly, the study reveals that solitary locusts exhibit a higher capacity for parthenogenesis compared to gregarious ones. However, parthenogenesis results in significantly lower hatching rates than sexual reproduction. These findings contribute to our understanding of reproductive strategies in locusts and raise thought-provoking questions regarding the evolutionary ecology of parthenogenesis. I suggest revisions to improve the clarity of result presentation, statistical analysis, and discussion structure. Please find detailed comments below.

Major Comments:

1. Introduction:

  • I recommend that the authors clearly articulate the central scientific question and summarize the experimental design in the final paragraph of the Introduction. This will help readers understand the scope and rationale of the study.

2. Results:

  • The results section is somewhat difficult to follow due to the order and logic of comparisons. The study seems to contain three major comparisons:

    1. B0 vs. GG: To assess the effect of reproductive mode (parthenogenetic vs. sexual).

    2. SS vs. GG: To examine the effect of rearing density throughout development on parthenogenesis.

    3. GG vs. GSH, GSL, GSM: To isolate the effect of density experienced during the nymphal stage.

    I strongly suggest the authors reorganize the Results section according to this experimental structure. The comparison between B0 and SS should be excluded, as it conflates two variables: the presence of males and rearing density.

  • In Figure 3, the term "spawning stage" should be clearly defined, ideally in the Methods section.

  • The temporal oviposition patterns shown in the GSH, GSL, and GSM groups are intriguing. However, additional data or discussion on fecundity, hatching rates, and hatching dynamics in these groups would significantly enrich the results.

  • Statistical analysis needs more careful consideration:

    • For Figure 1 (and similarly Figure 4), the authors compare average pod numbers and note earlier peaks in GG groups. However, R² values reflect curve fitting, not statistical significance between groups. A better approach might be comparing the peak pod number per individual across groups using appropriate statistical tests.

    • In Figures 2a and 2c, the bar plots should represent all individual samples, not just group means, to allow meaningful statistical comparisons.

3. Discussion:

  • Lines 226–228: The sentence refers to “inconsistent results,” but this is not clearly supported or explained in the Results section. The study appears to control rather than quantify density. If inconsistencies are a key finding, please elaborate and link them clearly to the results from GSH, GSL, and GSM.

  • Lines 241–243: The concept of facultative parthenogenesis is important and should be introduced earlier in the paragraph for better logical flow.

  • Paragraph 2: This section on solitary locusts’ reproductive strategies is especially insightful. I recommend framing this discussion within behavioral ecology theories, such as terminal investment or bet-hedging strategies. It would also be valuable to explore whether similar patterns have been observed in other species.

  • Paragraph 3: While listing other cases of parthenogenesis is helpful, a stronger link should be drawn between these examples and the current study. How does this study advance our understanding of the evolutionary ecology of insect parthenogenesis? For instance, does life-history variation (e.g., lifespan, reproductive timing) correlate with parthenogenetic capacity? The lower hatching success observed raises the question: could there be a trade-off between hatching rate and species longevity?

Minor Comments:

  1. The manuscript would benefit from thorough language editing to correct grammatical errors, typos, and overly colloquial expressions (e.g., avoid informal phrases like “wonder”).

  2. Some parts of the text could be more concise. For example, Lines 140–143 could be moved to the Introduction or removed if redundant.

  3. Ensure consistent formatting throughout the manuscript:

    • Use spaces before and after mathematical or statistical symbols.

    • Leave a space between numbers and units (e.g., “10 mg” not “10mg”).

Author Response

Reviewer2:

This is a very interesting and valuable study. The authors investigate how population density influences parthenogenesis in locusts, a species known for striking density-dependent phenotypic plasticity. Surprisingly, the study reveals that solitary locusts exhibit a higher capacity for parthenogenesis compared to gregarious ones. However, parthenogenesis results in significantly lower hatching rates than sexual reproduction. These findings contribute to our understanding of reproductive strategies in locusts and raise thought-provoking questions regarding the evolutionary ecology of parthenogenesis. I suggest revisions to improve the clarity of result presentation, statistical analysis, and discussion structure. Please find detailed comments below.

Response:

We thank the reviewer for these comments.

I recommend that the authors clearly articulate the central scientific question and summarize the experimental design in the final paragraph of the Introduction. This will help readers understand the scope and rationale of the study.

Response:

Thanks. We followed your advice and added the central scientific question after the introduction, as well as the experimental design in question.

The results section is somewhat difficult to follow due to the order and logic of comparisons. The study seems to contain three major comparisons:

B0 vs. GG: To assess the effect of reproductive mode (parthenogenetic vs. sexual).

SS vs. GG: To examine the effect of rearing density throughout development on parthenogenesis.

GG vs. GSH, GSL, GSM: To isolate the effect of density experienced during the nymphal stage.

I strongly suggest the authors reorganize the Results section according to this experimental structure. The comparison between B0 and SS should be excluded, as it conflates two variables: the presence of males and rearing density.

Response:

We appreciate the reviewer’s insightful analysis. So, we have cancelled the comparison between B0 and SS in the manuscript now. We did not organize the Results section based on the experimental structure, but on the sequence of the oviposition process, including oviposition dynamics, fecundity, hatching rate and hatching dynamics. To facilitate understanding of the setup of our experimental group, we have added Table 1 into the manuscript.

Table 1. Rearing conditions for each experimental group.

Group

Nymphs

Adults

Reproduction

SS

Solitary individual

Solitary individual

Unfertilized

GG

Gregarious

Gregarious

Unfertilized

B0

Gregarious

Gregarious

Fertilized

GSL

Low-density Gregarious

Solitary individual

Unfertilized

GSM

Medium-density Gregarious

Solitary individual

Unfertilized

GSH

High-density Gregarious

Solitary individual

Unfertilized

In Figure 3, the term "spawning stage" should be clearly defined, ideally in the Methods section.

Response:

Thanks. We followed your advice and added the definitions in “spawning stage”

The temporal oviposition patterns shown in the GSH, GSL, and GSM groups are intriguing. However, additional data or discussion on fecundity, hatching rates, and hatching dynamics in these groups would significantly enrich the results.)

Response:

Thanks. The three groups had relatively low numbers of oviposition events, with most individuals failing to oviposit throughout their lifespans. The limited oviposition events were concentrated among a small number of individuals. Additionally, each pod contained a very small number of eggs, and the eggs produced showed almost no hatching. These phenomena will be the focus of future research and are therefore not presented here.

Statistical analysis needs more careful consideration:

For Figure 1 (and similarly Figure 4), the authors compare average pod numbers and note earlier peaks in GG groups. However, R² values reflect curve fitting, not statistical significance between groups. A better approach might be comparing the peak pod number per individual across groups using appropriate statistical tests.

Response:

Thanks. Due to the group housing of the GG group, tracking the behavior of each individual became prohibitively difficult. Consequently, we substituted the overall average for individual-level analysis, instead of collection of each individual dataset. Therefore, we did not conduct statistical analysis between individuals, but only aimed to show the oviposition trend.

In Figures 2a and 2c, the bar plots should represent all individual samples, not just group means, to allow meaningful statistical comparisons.

Response:

Lines 226–228: The sentence refers to “inconsistent results,” but this is not clearly supported or explained in the Results section. The study appears to control rather than quantify density. If inconsistencies are a key finding, please elaborate and link them clearly to the results from GSH, GSL, and GSM.

Response:

Thanks. The discrepancy observed was initially due to our lack of awareness regarding the impact of larval density on the behavior of adults. Upon post-analysis review, we identified a potential correlation between the two, leading us to design this experiment for verification. The differences have been demonstrated in this article.

Lines 241–243: The concept of facultative parthenogenesis is important and should be introduced earlier in the paragraph for better logical flow.

Response:

Thanks. We followed your advice and make the concept of facultative parthenogenesis introduced earlier in the paragraph.

Paragraph 2: This section on solitary locusts’ reproductive strategies is especially insightful. I recommend framing this discussion within behavioral ecology theories, such as terminal investment or bet-hedging strategies. It would also be valuable to explore whether similar patterns have been observed in other species.

Response:

Thanks. We followed your advice and discussed our study within behavioral ecology theories.

Paragraph 3: While listing other cases of parthenogenesis is helpful, a stronger link should be drawn between these examples and the current study. How does this study advance our understanding of the evolutionary ecology of insect parthenogenesis? For instance, does life-history variation (e.g., lifespan, reproductive timing) correlate with parthenogenetic capacity? The lower hatching success observed raises the question: could there be a trade-off between hatching rate and species longevity?

Response:

Thanks. We strengthened the relationship between the cases cited and our research, and conducted a more in-depth analysis.

Minor Comments:

The manuscript would benefit from thorough language editing to correct grammatical errors, typos, and overly colloquial expressions (e.g., avoid informal phrases like “wonder”).

Some parts of the text could be more concise. For example, Lines 140–143 could be moved to the Introduction or removed if redundant.

Ensure consistent formatting throughout the manuscript:

Use spaces before and after mathematical or statistical symbols.

Leave a space between numbers and units (e.g., “10 mg” not “10mg”).

Response:

Thanks. We followed your advice and adjusted the format of the manuscript.

Reviewer 3 Report

Comments and Suggestions for Authors

A most interesting study of a little studied phenomenon, so with some minor corrections, this paper is well worth publishing.  See attached comments

Author Response

Reviewer3:

A most interesting study of a little studied phenomenon, so with some minor corrections, this paper is well worth publishing.

Response:

We thank the reviewer for these comments.

Line 58 : ‘offpspring’ NOT ‘offsprings’

Line 71 : “…University as part of a long-term laboratory colony.”

Lines 90, 95 and 105-6: You do not need to mention again that they were fed: Line 82 mentions daily feeding and everyone will assume locusts were also fed in each experiment.

Line 115: “At 5-7 days after eclosion, when the locusts were reaching sexual maturity,,..”

Line 125: “Severely moldy eggs were omitted from the results.”

Response:

Thank you for your careful reading and optimization of the manuscript's language. We have adopted your suggestions

I suggest a table outlining the rearing conditions for each group, so that the reader can quickly determine what GG, SS, B0, GSL etc mean when looking at the results:

Table 1. Rearing conditions for each experimental group

Group                     Nymphs                        Adults

SS                  solitary individual solitary          individual unfertilized

GG                       gregarious                 gregarious unfertilized

B0                       gregarious                  gregarious fertilized

GSL                  low-density gregarious        solitary individual unfertilized

GSM               medium-density gregarious       solitary individual unfertilized

GSH                 high-density gregarious        solitary individual unfertilized

Response:

Thank you for your constructive suggestion. We have added a table to the manuscript as shown in the table below.

Table 1. Rearing conditions for each experimental group

Group

Nymphs

Adults

Reproduction

SS

Solitary individual

Solitary individual

Unfertilized

GG

Gregarious

Gregarious

Unfertilized

B0

Gregarious

Gregarious

Fertilized

GSL

Low-density Gregarious

Solitary individual

Unfertilized

GSM

Medium-density Gregarious

Solitary individual

Unfertilized

GSH

High-density Gregarious

Solitary individual

Unfertilized

Fig 1 had me completely confused at first. That is because GG is blue in Fig 1a and black in Fig 1b. Make GG blue in both so that the reader immediately recognizes that the shape of the graph for GG is the same in Fig 1a and Fig 1b. To make comparison of Fig 1a and 1b easier, make length of the “Oviposition day” the same: have both 1a and 1b go to 50 with the same tick marks.

Response:

Thanks, and we followed your advice and adjusted Fig 1.

Line 149: Saying “both GG groups” is confusing: say “GG group peaked earlier and laid the least number of egg pods, while SS laid the most.”

Response:

Thanks, and we followed your advice and adjusted our manuscript.

And add more about the differences between GSL, GSM and GSH: Line 152-158: “…parthenogenesis dynamics. When gregarious nymphs were separated into three density levels, peak laying by the resulting unfertilized females concentrated around 15 days, but lower density gregarious nymphs led to laying over a longer time period, with more egg pods than with any of the others including GG (Fig 1b: GSL). Medium-density gregarious nymphs led laying over a shorter period, while high-density gregarious nymphs led to laying of only a few egg pods over a very short period (Fig 1b).

Response:

Thanks, and we we followed your advice and adjusted our manuscript.

Lines 166-182: “Solitary females (SS) laid the highest number of egg pods (Fig 2a) but had the least number of eggs per pod (Fig 2b). In contrast, fertilized gregarious females (B0) laid fewer pods (Fig 2a) but had the most eggs per pod (Fig 2b). The 2 overall result was that SS and B0 laid a similar number of eggs (Fig 2c). Gregarious females that were unfertilized (GG) laid only a few pods (Fig 2a). Some of the pods had a similar number of eggs per pod as the average for fertilized gregarious females (B0), while others had very few eggs (Fig 2b), resulting in far fewer eggs per parthogenetic female (Fig 2c). Thus, density level…”

Response:

Thanks, and we followed your advice and adjusted our manuscript.

Fig 3 & 4: I think “oviposition” as used in Figure 1 is better than “spawning”. Line 215: “The hatching peaks of the B0 and SS groups were similar, with the GG group slightly delayed (Fig. 4). However, the peak values….” Omit lines 218-221 because while B0 does start earlier than SS, both have peak hatching around 15-16 days. For both B0 and SS, hatching is a normal distribution about the mean: so that both have a mean + standard error. B0 may have a slightly larger S.E. so that hatching begins earlier and continues later, but it may just be that there are more hatchings, so some begin earlier.

Response:

Thanks, and we followed your advice and replaced “spawning” with “oviposition”. Additionally, we mentioned your analysis in the discussion section.

Discussion, Lines 226-ff: a good point to make that the effects of gregariousness are complex requiring several kinds of experiments to elucidate.

Response:

Thanks, and we followed your advice and discussed the effects of gregariousness in the discussion section.

Line 236: “..thus resorting to more frequent but smaller-scale…”

Line 246: “…eggs increased for eggs laid late in the oviposition period. This pattern may be part of making certain some successful reproduction occurs through investing more energy into parthenogenic reproduction in later stages when the females are getting older and nearer death. We found that solitary-phase locusts typically die soon (within 3.5 days) after their final oviposition event.

Response:

Thanks. We have adopted your suggestions and incorporated them into the article.

Line 316-17: the non-significance with gregarious locusts may reflect the far fewer

eggs hatching.

Response:

Thanks. We followed your advice and adjusted our manuscript.

Round 2

Reviewer 2 Report

Comments and Suggestions for Authors

The authors have made substantial improvements to this manuscript. I still have a few minor comments:

  1. I have not seen a response to my earlier comment: "In Figures 2a and 2c, the bar plots should represent all individual samples, not just group means, to allow meaningful statistical comparisons."

  2. For the visualization in Figure 3d, I suggest reducing the range of the Y-axis so that the columns appear longer and the differences between groups are more apparent.

  3. Please carefully revisit the minor comments provided in the previous round. There are still some typographical and language issues (e.g., Line 172: the word “Wonder” should be revised; ensure spaces are used before and after mathematical or statistical symbols).

Author Response

Dear reviewer, We thank you for your efforts in commenting our manuscript. We have gone through all comments and suggestions (all changes indicated by blue color in the main text) and hope that our changes and answers fulfil your expectations. Reviewer Comments I have not seen a response to my earlier comment: "In Figures 2a and 2c, the bar plots should represent all individual samples, not just group means, to allow meaningful statistical comparisons." Response: Thanks. We have used the overall average for individual-level analysis, instead of collection of each individual dataset in GG group, because of difficulty in tracking the oviposition of each individual. And we have added more explanation about the Y axis labeling in Materials and Methods for clearer understanding. For the visualization in Figure 3d, I suggest reducing the range of the Y-axis so that the columns appear longer and the differences between groups are more apparent. Response: Thanks. We have adjusted Figure 3d according to your suggestions to reduce the range of the Y-axis. Please carefully revisit the minor comments provided in the previous round. There are still some typographical and language issues (e.g., Line 172: the word “Wonder” should be revised; ensure spaces are used before and after mathematical or statistical symbols). Response: Thanks. We adjusted all the wording carefully by checking the entire text.